# Temperature-Compensated Multi-Point Strain Sensing Based on Cascaded FBG and Optical FMCW Interferometry

**DOI:** 10.3390/s22113970

**Published:** 2022-05-24

**Authors:** Zhiyu Feng, Yu Cheng, Ming Chen, Libo Yuan, Deng Hong, Litong Li

**Affiliations:** 1State Key Laboratory of Optical Fiber and Cable Manufacture Technology, Yangtze Optical Fiber and Cable Joint Stock Limited Company, Wuhan 430073, China; 20082304017@mails.guet.edu.cn (Z.F.); hongdeng@yofc.com (D.H.); lilitong@yofc.com (L.L.); 2College of Photonic and Electronic Engineering, Guilin University of Electronic Technology, Guilin 541004, China; mchen@guet.edu.cn (M.C.); lbyuan@guet.edu.cn (L.Y.); 3Guangxi Key Laboratory of Optoelectronic Information Processing, Guilin 541004, China

**Keywords:** optical FMCW interferometry, cascaded FBG, back Rayleigh scattering, strain sensing, temperature compensation

## Abstract

We proposed a novel temperature-compensated multi-point strain sensing system based on cascaded FBG and optical FMCW interferometry. The former is used for simultaneous sensing of temperature and strain, and the latter is used for position information reading and multiplexing. In the experiment, a narrow linewidth laser with continuous frequency-sweeping was used as the light source. After demodulating the beat-frequency signal, the link information of the 16 m fiber was obtained, and the measured result was identical to the actual position. The measurement accuracy reached 50.15 mm, and the dynamic range was up to 22.68 dB. Meanwhile, we completed the sensing experiments for temperature range from 20 °C to 90 °C and strain range from 0 με to 7000 με. The sensitivity of the sensing system to temperature was 10.21 pm/°C, the sensitivity and accuracy to strain were as high as 1.163 pm/με and 10 με, respectively. Finally, the measured strain and temperature values were obtained using the sensing matrix. The sensing system has important practical significance in the field of quasi-distributed strain measurement.

## 1. Introduction

Strain measurement plays an important role in structural health monitoring (SHM) such as aviation, ship engineering, and bridge construction. In recent years, optical fiber strain sensors have stood out among many sensing technologies due to their excellent properties such as anti-electromagnetic and radiation resistance, miniaturization, and high temperature and corrosion resistance, and have great development prospects. In terms of sensing principle, the fiber optic strain sensors proposed so far can be divided into three types. The first type is an interferometric fiber optic strain sensor, such as an all-fiber optical demodulator, based on the Michelson interferometer (MI) for signal interrogation [1]; ultra-sensitivity strain [2] and high-temperature strain [3] measurement are realized based on the Fabry-Perot interferometer (FPI), a hybrid interferometer that can measure strain and temperature simultaneously, composed of FPI and MI cascade with each other [4], and strain sensors based on the Mach-Zehnder interferometer(MZI) [5,6,7], etc. The second type is a fiber Bragg grating strain sensor, which is mainly used for structural health monitoring in the engineering field [8]. For example, fiber grating combined with the lever principle to achieve micro-strain measurement [9], a long-gauge fiber Bragg grating (FBG) strain sensor with enhanced strain sensitivity [10], or a silver-coated, 45-degree-radiated tilted fiber used for strain sensing [11], using three-dimensional (3D) printing technology to embed the FBG into a thermoplastic polyurethane filament can be effective in strain measurement [12], etc. The third type is the optical fiber surface plasmon resonance (SPR) strain sensor [13,14,15,16], which mainly makes the transmission light with total reflection at the fiber core and cladding interface contact with the metal film. However, the processing of this sensor is very difficult, and its repeatability is low. Among the above-mentioned sensors, using fiber Bragg grating (FBG) to measure strain has high sensitivity and fast response time; this, coupled with its strong reusability and low cost, means it is more likely to be favored by the engineering field. However, while using FBG to measure strain, the center wavelength of the grating is easily affected by the external ambient temperature. In order to improve the measurement accuracy of strain, the separation of temperature and strain is achieved by cascading FBG, so as to achieve strain measurement with temperature compensation.

However, FBGs can only be measured at a specified location. As the number of FBG multiplexes increases, a wide spectral range of light sources is required, and the exact location of the strain is not clearly distinguishable. The frequency-modulated continuous wave (FMCW) interferometry technique from the radar field can solve our problems. Optical FMCW interference technology, also known as optical frequency domain reflectometry (OFDR) technology, is a technique that detects backward Rayleigh scattering to obtain position information on an optical fiber link. OFDR is widely used in distributed sensing [17,18,19,20,21]. In recent years, a number of researchers have used a combination of FBG and OFDR techniques for strain measurements. Examples include the measurement of strain on an aircraft structure using optical fibers embedded with high-density FBGs [22], and the detection of spatial strain distribution within a concrete member using two vertical FBGs [23]. However, the sensors in the above case are not temperature compensated. In addition, some researchers have used polarization-maintaining FBGs (Panda-FBGs) [24], a large-multiplexing-capacity dense ultra-short (DUS)–FBG array [25], and tapered FBG (TFBG) [26] to achieve simultaneous temperature and strain measurements. However, in these studies with temperature compensation, the sensor requires a sophisticated fabrication process, which is very high cost and lacks the ability of multi-point measurement.

In this paper, we propose a novel multi-point strain sensing system with temperature compensation. Firstly, we establish a theoretical model of the sensing system, analyze the technical principles involved, and study the algorithms for signal demodulation. Secondly, an optical FMCW interferometer is built by using a narrow-linewidth laser with high-frequency sweep linearity as the light source, combined with cascaded FBG. Thirdly, after processing the beat-frequency interference signal obtained by the data acquisition card (DAQ), such as FFT, the link information of the 16 m optical fiber is obtained. The measured results correspond to the positions in the actual system. The measurement accuracy reaches 50.15 mm, and the dynamic range is up to 22.68 dB. Finally, we test the ability to simultaneously sense temperature and strain at a temperature range from 20 °C to 90 °C and a strain range from 0 με to 7000 με. The sensitivity of the sensor to temperature and strain is acquired up to 10.21 pm/°C and 1.163 pm/με, respectively. In addition, the strain measurement accuracy of 10 με is achieved by 20 strain repetitions. This allows us to evaluate the stability and accuracy of strain measurement. We obtained the measured strain and temperature values through the sensing matrix. Thus, the sensing system using optical FMCW interferometry combined with the cascaded FBG successfully monitors the axial strain distribution and implements the temperature compensation function. The sensing system has important practical significance in the field of quasi-distributed strain measurement.

## 2. Principle

The multi-point sensing system proposed in this paper is based on the combination of two technologies: optical FMCW interferometry and FBG. The former is used as an interrogation method, and the latter is used for external parameter sensing, such as strain and temperature sensing.

### 2.1. Optical FMCW Inteferometry

The optical FMCW interferometric technique used in this paper is realized based on the MZI, as shown in Figure 1. We used a tunable laser resource (TLS), and the tuning of the optical frequency was linear in the ideal state. The optical signal from the TLS was divided into two branches via optical coupler (OC) 1: one beam enters the reference arm and connects to the polarization controller (PC), as an intrinsic reference light; the other branch enters the test arm. Due to the presence of Rayleigh scattering and Fresnel reflection on the fiber under test (FUT), a part of the light is returned to the OC2 as test light. Two beam signals interfere at the OC2 with beat frequencies. Since the optical frequencies carried by the two beam signals are different, it is also called differential frequency interference. The beat-frequency signals are converted from optical signals to electrical signals by a photodetector (PD), and then captured by a DAQ. If the phase noise of the laser is negligible, the fast Fourier transmission (FFT) can be performed on the signal to convert it to the spatial domain. Thus, the position information on the FUT can be acquired.

The light signal output from the reference arm can be expressed by [20,27]
(1)$$E_{r}\left(t\right)=E_{0}\exp\left\{j\left[2\pi f_{0}t+\pi \gamma t^{2}+\phi \left(t\right)\right]\right\}$$
where $E_{0}$ is the initial amplitude of the output light signal, $f_{0}$ is the initial frequency, γ is the linear tuning speed of the light source, ϕ(t) is the initial phase, which represents the phase of the random fluctuations of the light source at time t.

Suppose there is a reflection point at *z* on the signal arm with a time delay $\tau_{Z}$ relative to the intrinsic reference light
(2)$$\tau_{Z}=2nz/c$$
where n is the effective refractive index of the fiber, z is the optical range difference between the signal arm at *z* and the reference arm, and c is the speed of light propagation in vacuum. Then the optical signal output from the signal arm is
(3)$$E_{r}\left(t\right)=\sqrt{R\left(\tau_{Z}\right)}E_{0}\exp\left\{j\left[2\pi f_{0}\left(t-\tau_{Z}\right)+\pi \gamma {\left(t-\tau_{Z}\right)}^{2}+\phi \left(t-\tau_{Z}\right)\right]\right\}$$
where $R\left(\tau_{Z}\right)$ is the reflection coefficient at the reflection point. Therefore, the beat-frequency interference signal generated by the reference arm and the signal arm at OC2 can be expressed as
(4)$$I\left(t\right)=2\sqrt{R\left(\tau_{Z}\right)}E_{0}{}^{2}\cos\left\{2\pi \left[f_{0}\tau_{Z}+f_{b}t-\frac{1}{2}\gamma \tau_{Z}{}^{2}+\phi \left(t\right)-\phi \left(t-\tau_{Z}\right)\right]\right\}$$
where $f_{b}$ is the beat frequency. As can be seen from the above equation, the phase of the beat-frequency signal mainly consists of three parts: the first part is the constant term $f_{0}\tau_{Z}-\frac{1}{2}\gamma \tau_{Z}{}^{2}$; the second part is the phase term that varies linearly with time, which is the beat frequency part $f_{b}t$; the third part is the phase term that varies nonlinearly with time $\phi \left(t\right)-\phi \left(t-\tau_{Z}\right)$, which is also called phase noise, and its presence deteriorates the spatial resolution of the system.

### 2.2. Principle of Grating Sensing

Generally speaking, FBG is formed by periodically modifying the refractive index of the fiber core along the fiber. According to the mode coupling theory, the mode coupling between two beams transmitted in reverse will occur at the resonant wavelength of the FBG, i.e., Bragg wavelength ($\lambda_{Bragg}$)
(5)$$\lambda_{Bragg}=2n_{eff}\Lambda$$
where $n_{eff}$ is the effective refractive index of the fiber core and Λ is the spatial period of the FBG. From the above equation, it can be seen that the variation of Bragg wavelength is only proportional to the effective refractive index and the spatial period. When the external environment (e.g., temperature and strain) changes, it changes the effective refractive index and the period of the grating, which causes the resonant wavelength of the grating to be shifted. The corresponding wavelength shifts can be expressed as [28]
(6)$$\frac{∆\lambda }{\lambda }=\left(1-\rho_{e}\right)\Delta \varepsilon +\left(\alpha_{f}+\xi_{f}\right)\Delta T$$
where Δε represents the axial strain on the fiber, ΔT represents the temperature change, $\rho_{e}$ is the elasticity coefficient of the fiber, and $\alpha_{f}$ and $\xi_{f}$ represent the thermo-optical and thermal expansion coefficients of the fiber, respectively.

It is worth noting that FBG is used instead of the reflection point in the optical FMCW interference technique, the backscattering returned by the circulator is enhanced.

### 2.3. Signal Demodulation Process

The reflected light returned by the FBGs on the test fiber have different time delays, $\tau_{Z}$, relative to reference light. When it interferes with the light from the reference branch, beat signals with different frequencies will be generated. These beat signals are converted by PD and acquired by DAQ. The TLS scans continuously, so the beat signals collected by the DAQ are mixed in the time domain (wavelength domain). By FFT algorithm, the information of each reflection point on the optical fiber link is separated in the frequency domain (spatial domain). After bandpass filtering and moving average processing, noise signal is reduced. Then, using the inverse fast Fourier transform (IFFT) algorithm [20,29], the peaks of interest are extracted and converted into time domain (spatial domain) so that the wavelength of each grating is demodulated out. The equal relation between time domain and wavelength domain can be obtained according to the formula
(7)$$∆T=\lambda_{range}/\left(N\cdot \gamma \right)$$
where ∆T is the sampling interval of the DAQ, $\lambda_{range}$ is the wavelength scanning range of TLS, and *N* is the sampling number.

By substituting $f_{b}=\gamma \tau_{Z}$ into Equation (2), the equal relation between the frequency domain and the spatial domain can be obtained
(8)$$f_{b}=2nz\gamma /c$$

## 3. Experimental Setup and Results

### 3.1. Experiment Setup

The system diagram is shown in Figure 2. Part I: TLS. The laser provides a continuous sweep signal for the system, which is used to generate the beat-frequency signal; in addition, it also provides a trigger signal to the DAQ for synchronization. Part II: Auxiliary interferometer. This part is based on the MZI. It consists of two branches of different lengths, which are used to obtain the real-time phase of the TLS and thus compensate for the phase spreading caused by nonlinear phase noise. Part III: Main interferometer, which is also based on the MZI configuration: one branch reduces the polarization effect through a PC; the other branch obtains the reflected back Rayleigh scattering and Fresnel reflection through the circulator. The two branches generate interference signals at the OC5, which are converted into electrical signals by the balanced photodetector (BPD). Part IV: Multiplexed apparatus, which consists of two OCs and three groups of sensors 1–3. Each set of sensors contains two FBGs, the first one is used for temperature-sensing and the second is for strain. The FBGs for temperature sensing are encapsulated in a protective tube of rigid material to isolate it from the effects of external strain. In contrast, FBGs used for strain sensing are directly exposed. Since each group of sensors is in parallel, in order to more accurately demodulate their position information, it is necessary to ensure that the Rayleigh backscattering returned by them is different, and the resulting beat signal frequency is inconsistent. Therefore, the distance between each group of sensors and the OC6 should be different. Part V: Signal acquisition and processing device. The DAQ acquires the signals from the main interferometer and auxiliary interferometer, and processes the signals with FFT and filtering to obtain the position domain information.

In addition, the information of the equipment is as follows: the TLS was from Luna Technologies (Phoenix 1400), the wavelength-sweeping range was from 1515 to 1565 nm, and sweeping speed is 40.01 nm/s. The output power of the TLS is 8 mW and the linewidth was 1.5 MHz. The BPD, which has a bandwidth of 400 MHz, was from Thorlabs (PDB570C). The DAQ was from National Instruments (PXIe 5114), the sampling frequency of which was 4.55 MS/s, corresponding to a sampling interval of 20 ms. According to the Nyquist–Shannon sampling theorem, the sampling frequency must be greater than twice the highest frequency of the signal under test. However, in practice, the sampling rate is generally chosen to be greater than 5~10 times the measured signal, so that the restored signal is closer to the original signal. The fiber in this experiment was a standard telecom single-mode fiber (YOFC Co., Ltd., Wuhan, China). A thermotank (DHG9030A) was used to simulate the external ambient temperature state. The air inside the tank was blown by wind to ensure a constant temperature inside it. The system used a micrometer driven fiber optic slide to precisely induce strain in the fiber, and the fiber-optic sliding table was from Zolix (NFP-3561L).

### 3.2. Fiber Optic Link Information Detection

The computer demodulated and analyzed the data acquired by the DAQ. The original beat-frequency signal is shown in Figure 3a below. The beat-frequency signal is not a regular signal, but a signal with many burrs. This is due to the nonlinear phase of the laser sweep, coupled with multiple reflection events on the sensing fiber, resulting in the superposition of multiple beat signals. The power spectrum is calculated by applying FFT on the original beat frequency signal, which leads to the position domain information, as shown in Figure 3b. The grating position information obtained after demodulation is consistent with the actual system, and the position information is shown in Table 1.

Apparently, there are multiple peaks in the power spectrum corresponding to the connector, the fusion point, sensor group 1 (containing FBG1 and FBG2), sensor group 2 (containing FBG3 and FBG4) and sensor group 3 (containing FBG5 and FBG6). The total length of the fiber is 16 m. The peak corresponding to the tail end is eliminated because antireflection treatment was applied to the tail end of the fiber. In addition to the large intensity peaks observed in the fiber link mentioned above, some low−intensity peak points were also detected. These are the Fresnel reflections in the optical fiber. As a whole, the dynamic range of the system is up to 22.68 dB, which makes the reflection events easy to observe.

Each group of sensors is composed of two FBGs, and their demodulation results are shown in Figure 3c–e. The positioning accuracy of the six sensing units is 5.15 cm and 13.07 cm, 18.90 cm and 18.08 cm, 16.60 cm and 17.89 cm, respectively. The reason for the fluctuation of the positioning accuracy is that the nonlinear phase of the beat frequency signal gradually increases with the increase in the distance. In addition, fluctuations in the power of the light source and the polarization effect also lead to the deterioration of the positioning accuracy.

Observing the two gratings in each group of sensors, we find that the intensity of the two units is not the same, and the intensity of the first grating is higher than that of the second grating. This is mainly because the grating is equivalent to a reflector, which will reflect back part of the transmitted light. The light signal through the first grating continues to propagate to the second grating. The intensity of light propagated to the second grating will be smaller than that propagated to the first grating, so the intensity of the beat signal generated by the two gratings will be different.

### 3.3. Temperature Compensation Experiment of Sensing System

Once the position information of the fiber link is obtained, the wavelength information of the grating is demodulated by the IFFT algorithm. To ensure that the strain sensing unit in each sensor group is not affected by the temperature sensing unit, it is necessary to ensure that the center wavelength of two units in the same sensing channel are not the same (e.g., FBG1 and FBG2). But the central wavelength in different sensing channels can be the same (e.g., FBG2 and FBG4). This allows the number of strain sensing channels to be increased. In this experiment, the center wavelengths of the two units are 1535 nm and 1555 nm, respectively. The demodulated parameters of sensors are shown in Table 1.

Then, each set of sensors is placed in the thermotank. The sensitivity of the two sensing units to temperature is measured from 20 °C to 90 °C with a step of 10 °C. Figure 4 shows the wavelength information demodulated from beat signal (in the case of sensor group 1). In Figure 4, the X-axis represents wavelength information, and the Y-axis represents normalized intensity (Norm. I.). Both units are affected by the temperature and have the same trend. As the temperature increases, the central wavelength is red-shifted. Figure 5 shows the linearity of the sensing units in this system. The sensitivity to temperature is 10.21 pm/°C, 10.01 pm/°C, 9.97 pm/°C, 9.99 pm/°C, 10.01 pm/°C, and 10.11 pm/°C, respectively, with an average sensitivity of 10.05 pm/°C. This is consistent with the sensitivity of conventional FBG to temperature. Actually, the sensitivity of the six FBGs to temperature is slightly different, which is mainly determined by the reflection wavelength of the fiber grating and the natural properties of the glass fiber used in the fiber [8]. As shown in Figure 5, it can be seen that the correlation linear coefficient, R^2^, of all the sensing units is greater than 0.9994, and FBG3 has the largest linear correlation coefficient of 0.99995.

### 3.4. Multi-Point Strain Experiment for Sensing System

Next, we mounted a fiber-optic sliding table (FST) with a 10 μm resolution micrometer driver on an aluminum rail. The strain sensing units were placed on it and, after keeping one end of each unit in a free condition (null strain state), the other ends were fixed by means of a clamp. The length of the units between the two ends was 100 mm. One of the ends was driven by a micrometer to move ∆*x*, and the ratio of this elongation, ∆*x*, to the total length, *L*, corresponds to the strain ∆*ε =* ∆*x*/*L*. The ambient temperature was kept constant at 20 °C and a strain range from 0 με to 7000 με was applied with the interval of 1000 με.

Figure 6 shows the experimental results of the sensor (in the case of sensor group 1) subjected to a series of strains. As shown in Figure 6a, the corresponding wavelength does not change because FBG1 is guaranteed to be unaffected by external strain. An enlarged view of the peak fluctuation is attached in Figure 6a, and the fluctuation is less than 5 pm. There are two main reasons for this fluctuation: on the one hand, the temperature will fluctuate slightly, which is related to the performance of the thermotank; on the other hand, due to the random error in the process of measurement and demodulation, the fluctuation of peak value is inevitable. Obviously, the central wavelength of FBG1 is at 1135.065 nm, which is consistent with Table 1. In contrast, the central wavelength of FBG2 is redshifted with increasing strain, and the shift is more pronounced relative to that caused by temperature change (in Figure 6b). Figure 7 illustrates the strain-sensing degree of the six sensing units. As strain-sensing units, the sensitivity of FBG2, FBG4 and FBG6 is 1.163 pm/με, 0.998 pm/με and 1.035 pm/με, respectively. The average sensitivity of them is 1.065 pm/με, whereas the temperature sensing units are not affected by external strain. The linear correlation coefficients of the three strain sensing units are all greater than 0.998, and the linear fitting degree of FBG2 is the highest, reaching 0.99989.

Finally, the temperature variation $∆T_{i}\left(i=1,2,3\ldots \ldots \right)$ and strain variation $∆\varepsilon_{i}\left(i=1,2,3\ldots \ldots \right)$ of the multiplexed sensor can be demodulated by a sensing matrix, which is shown below.
(9)$$\left[\begin{array}{c} ∆\varepsilon_{i}\\ ∆T_{i} \end{array}\right]=\frac{1}{S_{\varepsilon \_\varepsilon ,i}\cdot S_{T\_T,i}}\left[\begin{array}{cc} s_{T\_T,i} & -S_{\varepsilon \_T,i}\\ 0 & S_{\varepsilon \_\varepsilon ,i} \end{array}\right]\left[\begin{array}{c} ∆\lambda_{\varepsilon ,i}\\ ∆\lambda_{T,i} \end{array}\right],\;i=1,2,3\ldots \ldots$$
where $S_{\varepsilon \_\varepsilon ,i}$ is the sensitivity of the strain sensing units to strain in the ith sensor, $s_{T\_T,i}$ is the sensitivity of the temperature sensing units to temperature, $S_{\varepsilon \_T,i}$ is the sensitivity of the strain sensing units to temperature, $∆\lambda_{\varepsilon ,i}$ is the total central wavelength shift of the strain sensing units, and $∆\lambda_{T,i}$ is the amount of central wavelength shift of the temperature sensing units affected by temperature. Based on the above sensing matrix, we achieved the simultaneous measurement of strain and temperature.

### 3.5. Wavelength Stability of Strain Measurement System

To evaluate the stability and accuracy of the sensing system in measuring strain, we conducted 20 replicate experiments with sensor group 3 at a temperature of 20 °C and a strain of 5000 με. The data of each strain measurement was demodulated separately, as shown in Figure 8. The average wavelength of repeated strain measurement was 1560.235 nm. As shown in Figure 8c, the first data deviation is large. According to the *t*-test criterion, there is a large error in the first data. The root mean square (RMS) after elimination is 10.283 pm, which corresponds to an offset of 10 με, so the accuracy of the strain measurement can be determined as 10 με. In addition, the dispersion coefficient of 20 strain repeated measurements is 0.0012%, indicating that the sensing system is distributed with low difference.

## 4. Conclusions

In summary, we proposed a novel multipoint strain measurement system with temperature compensation, which is based on cascaded FBGs and optical FMCW interferometry. Theoretically, we analyzed the technical principles involved and studied the algorithms for signal demodulation. Experimentally, an optical FMCW interferometer was built using a narrow linewidth laser with high-sweep linearity as the light source, combined with a cascaded FBG. The link information of the 16-meter fiber was obtained by demodulating the beat frequency signal, and the results corresponded to the location in the actual system. The measurement accuracy reached 50.15 mm and the dynamic achieved 22.68 dB. We also tested the ability to simultaneously sense temperature and strain at a temperature range from 20 °C to 90 °C and a strain range from 0 με to 7000 με. The temperature and strain sensitivities of the three sets of cascaded FBGs is up to 10.21 pm/°C and 1.163 pm/με, respectively. In addition, the accuracy of the strain measurement was 10 με obtained from 20 replicate experiments. Therefore, the sensing system using optical FMCW interferometry combined with cascaded FBGs successfully monitors the axial strain distribution and also realizes the function of temperature compensation. The sensing system has important practical significance in the field of quasi-distributed strain measurement.

## Figures and Tables

**Figure 1 sensors-22-03970-f001:**
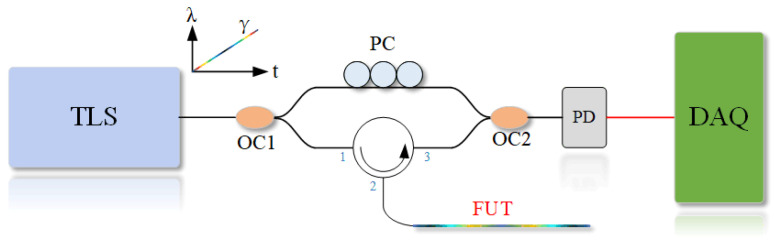
Schematic diagram of the sensing system based on optical FMCW interferometer. TLS: tunable laser source; OC: optical coupler; PC: polarization controller; PD: photodetector; DAQ: data acquisition card; FUT: fiber under test.

**Figure 2 sensors-22-03970-f002:**
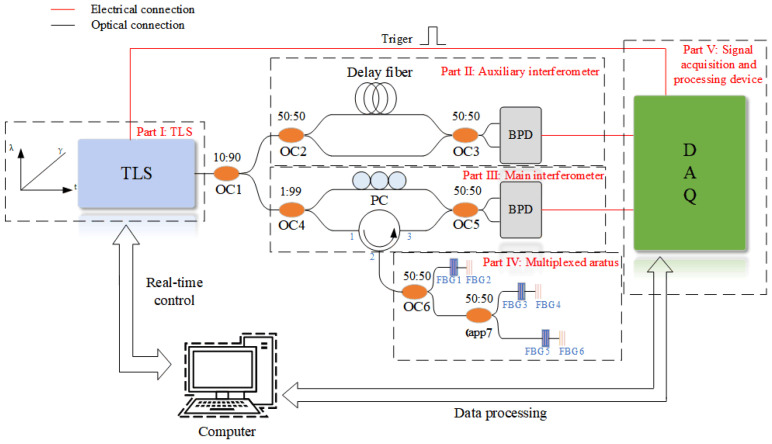
Multi-point strain measurement system with temperature compensation. TLS: tunable laser source; OC: optical coupler; BPD: balanced photodetector; PC: polarization controller; DAQ: data acquisition card.

**Figure 3 sensors-22-03970-f003:**
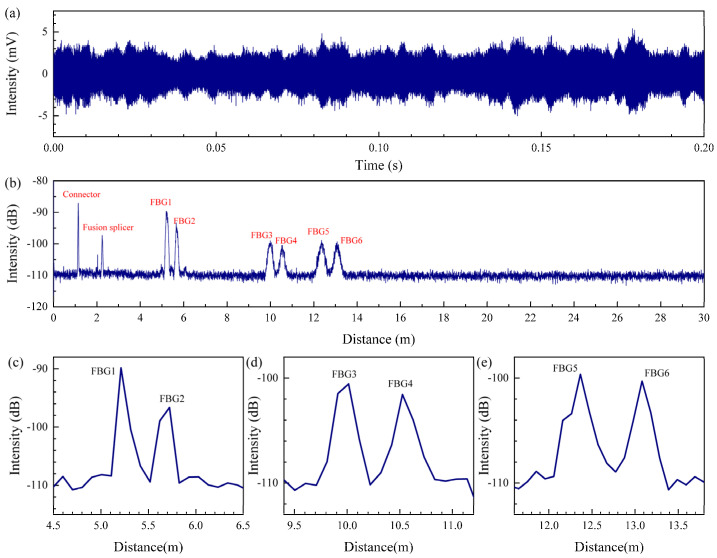
Demodulation process of beat frequency signal: (**a**) the original beat frequency signal; (**b**) location information corresponding to the beat frequency signal; (**c**) demodulation results for sensor group 1; (**d**) demodulation results for sensor group 2; (**e**) demodulation results for sensor group 3.

**Figure 4 sensors-22-03970-f004:**
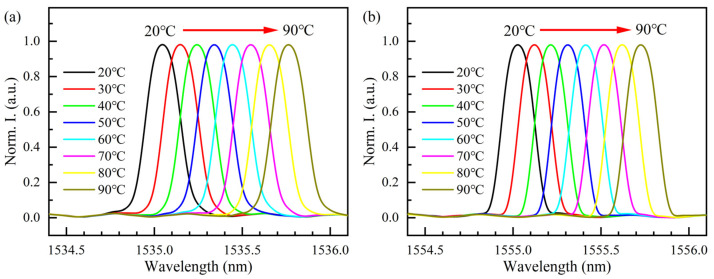
Normalized spectral reflectance of sensor group 1 under temperature experiment: (**a**) FBG1 at different temperatures; (**b**) FBG2 at different temperatures.

**Figure 5 sensors-22-03970-f005:**
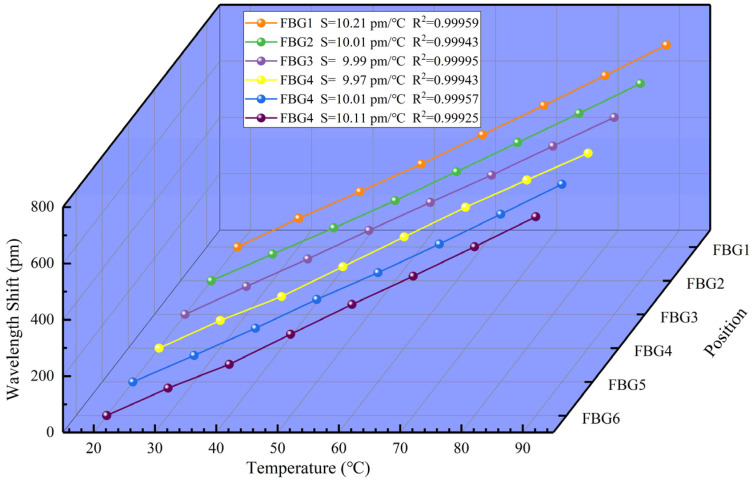
Temperature sensitivity of different sensing units.

**Figure 6 sensors-22-03970-f006:**
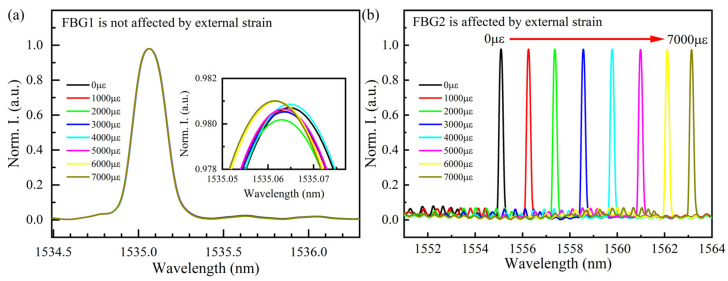
Normalized spectral reflectance of sensor group 1 under strain experiment: (**a**) FBG1 at different strains; (**b**) FBG2 at different strains.

**Figure 7 sensors-22-03970-f007:**
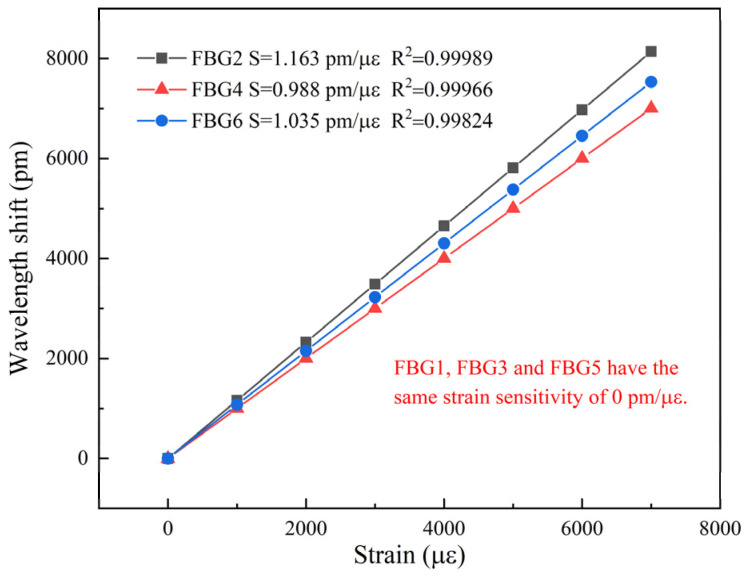
Sensitive performance of different sensing units to strain.

**Figure 8 sensors-22-03970-f008:**
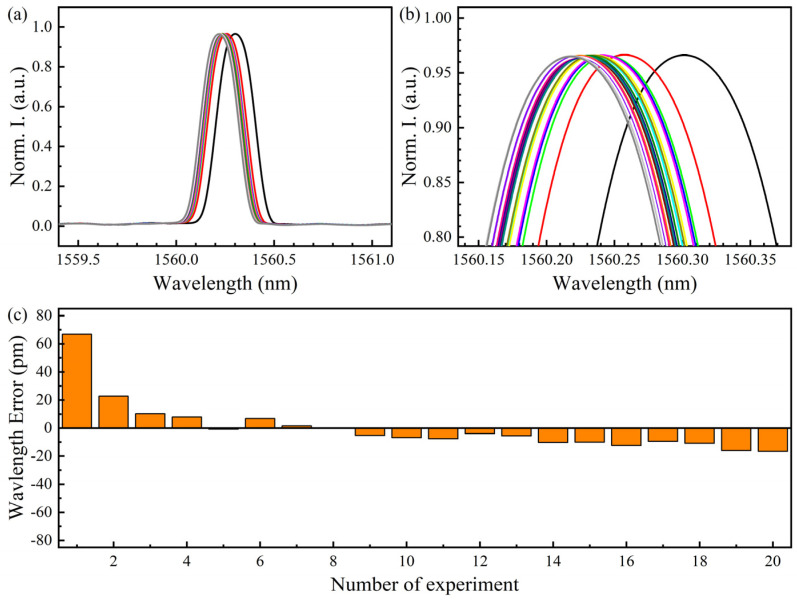
Results of repeated strain measurements: (**a**) normalized spectrum of the 20 times beat frequency signal; (**b**) close view of beat frequency signal; (**c**) the wavelength deviation of 20 repetitions.

**Table 1 sensors-22-03970-t001:** Basic parameters of multiplexed sensors.

Serial		Wavelength (nm)	Position (m)	3 dB Width ^1^ (cm)
Group 1	FBG1	1535	5.2112	5.15
	FBG2	1555	5.7221	13.07
Group 2	FBG3	1535	10.0136	18.90
	FBG4	1555	10.5245	18.08
Group 3	FBG5	1535	12.3638	16.60
	FBG6	1555	13.0752	17.89

^1^ 3 dB width represents positioning accuracy.

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
