# Peer review of "Temperature-Compensated Multi-Point Strain Sensing Based on Cascaded FBG and Optical FMCW Interferometry"

_sensors, 2022, doi:10.3390/s22113970_

Round 1

Reviewer 1 Report

The object of research described in the manuscript is an optical fiber sensor using Fiber Bragg Gratings (FBGs) and Frequency Modulated Continuous Wave (FMCW) optical interferometric detection. The sensor is used for strain measurement and it features temperature compensation based on FBGs dedicated to temperature measurement.

The use of FBGs for sensing has been the subject of research since at least 1990, with commercial sensing solutions being available from several vendors. Similarly, Frequency Modulated Continuous Wave (FMCW) optical interferometric techniques were first devised around that time, although they have not reached commercial success comparable to FBG-based systems. However, these areas of research are still of interest to several research groups.

There are some issues that must be addressed prior to publication of this manuscript.

First, Section 2 (Principle) should be rewritten to include a brief description of how the signal from FBGs is detected using FMCW interferometer described in Section 3. This is one of the key steps in the operation of the sensor, and as such it should be properly described. At present, the only mention of that process appears in Lines 145-149, referring the reader to Ref. 21.

Second, the term ‘Rayleigh scattering’ refers to scattering of optical radiation on particles whose size is much smaller that the wavelength of incident radiation. Therefore, it should not be used to describe any reflected radiation, as was done in Lines 225, 238 and 261.

Third, Captions of Figure 4 and 6 are incorrect. Most probably, they should read: ‘Normalized spectral reflectance of sensor group 1 grating’. Please consider adding information to Figure 6 that strain is applied only to FBG 2.

Fourth, results of repeated strain measurements (Figure 8) show a clear trend, that is not mentioned in Subsection 3.5. As there is no information in the manuscript about the fiber used in this experiment, one can only speculate that this trend results from creep taking place in the structure of the fiber, on the boundary between cladding and coating. Please add at least information about the fiber used in the experiment.

Fifth, the claim made in Lines 244-6 that temperature sensitivity of FBGs is related to their reflectivity, without providing any reference, is somewhat problematic.

Finally, there is a number of editorial type issues. Some imprecise and rather unfortunate expressions appear in the manuscript, e.g. ‘The upper computer demodulates and analyzes the data’ at Line 188. Please note that there is only one computer in Figure 2. Similarly,

‘because we processed the tail end of the fiber’ (Lines 203-4) can be expressed more accurately by ‘because antireflection coating/treatment was applied to the tail end of the fiber’. Expression ‘Distance … should be kept spatially independent’ should be rephrased.

Description of the measurement system in Subsection 3.1. refers to five parts of the systems. Unfortunately, these parts are not marked in Figure 2. Please consider marking them.

In conclusion, the manuscript cannot be published in its present form, before problems outlined in this review are addressed.

Author Response

Dear Reviewers,

The following is my reply to each of your comments, looking forward to your approval. We are truly grateful to you for your valuable advice. According to your suggestions, we have made careful modifications on the original manuscript. All changes made to the text are in red color. Attachment is our point-by-point responses to the reviewers’ comments. Please see the attachment.

Reviewer 2 Report

The authors proposed a novel temperature-compensated multi-point strain sensing system based on cascaded-FBG and optical FMCW interferometry. The results show that the system has good strain measurement accuracy and the theoretical calculation corresponds to the actual experiment. The paper makes a good contribution in terms of application in the field. Yet, some method and data descriptions are not clear in the manuscript. I recommend publication, if the following mandatory revisions are made well. Below are some specific comments:
1.    [Page1, lines 29 to 32], the authors should include more optical fiber strain sensors references.
2.    It is known that the temperature has a huge impact on the FBG operation. Is there any difference between the actual temperature and the measurement results at constant temperature? In addition, additional performance verification studies for heating and cooling processes are recommended. And when suffering from inhomogeneous temperatures and strain, the spectrum of the FBG sensors may be distorted or overlapping. How would the authors' proposed sensing system perform under these conditions?
3.    Abstract- [Page1, line19] ….sensing experiments for temperature range from 30°C to 90°C and strain range from 0 με to 7000 με. In the text- [Page8, line283-285]. Wavelength stability of strain measurement system, we conducted 20 replicate experiments at a temperature of 20°C and a strain of 5000 με. Why is the stability of the measurement system measured at 20°C and not at temperatures in the temperature range 30°C to 90°C.    
4.    Figure 5. shows the linearity of the sensing units in this system, it is recommended to increase the value of the correlation linear coefficient (r^2) for comparative analysis.
5.    Figure. 6 (a) has weak quality. (graphic overlapping is not easy to distinguish) Please improve.
6.    Figure 7. Sensitive performance of different sensing units to strain. FBG1, FBG3 and FBG5 are not clearly labeled and not easily distinguished in the figure. Please improve.
7.    Figure 8 shows the results of 20 repeated strain measurements. In Figures (a) and (b), there is a wavelength shift that is too large, and Figure (c) has the largest wavelength error in the first experiment. It is suggested that the author use statistical t test to explain. Also, In the text- [Page8, line283-285], what is the coefficient of variation (CV.%) for the repeatability of 20 replicate strain measurements?
8.    What are the future directions of this research? Authors are encouraged to write the future direction.

Author Response

(The authors gave the same response as above.)

Round 2

Reviewer 1 Report

The issues listed in my previous review were addressed properly. There are two minor issues that should be addressed.

First, from the figure caption under Fig. 6a follows that strain applied to FBG1 does not cause any change in wavelength.  This most probably means that FBG1 is installed in a protective housing. Please consider including this information in the description of the measurement setup.

 Second, an imprecise and rather unfortunate expression appears at lines 159-160 ‘When they interfere with another branch’. Most probably, it should read ‘When it interferes with light reflected from the other branch’.

In conclusion, the manuscript can be published in its present form.

Author Response

Dear Reviewer,

The following is my reply to each of your comments, looking forward to your approval. We are truly grateful to you for your valuable advice. According to your suggestions, we have made careful modifications on the original manuscript. All changes made to the text are in red color. Attachment is our point-by-point responses to the your comments. Please see the attachment.

Yours sincerely,

Zhiyu Feng

2022.5.21
